# Mutation bias interacts with composition bias to influence adaptive evolution

**Alejandro V. Cano** [1,2], **Joshua L. Payne** [1,2]*

**1** Institute of Integrative Biology, ETH, Zurich, Switzerland, **2** Swiss Institute of Bioinformatics, Lausanne, Switzerland

* joshua.payne@env.ethz.ch

## Abstract

Mutation is a biased stochastic process, with some types of mutations occurring more frequently than others. Previous work has used synthetic genotype-phenotype landscapes to study how such mutation bias affects adaptive evolution. Here, we consider 746 empirical genotype-phenotype landscapes, each of which describes the binding affinity of target DNA sequences to a transcription factor, to study the influence of mutation bias on adaptive evolution of increased binding affinity. By using empirical genotype-phenotype landscapes, we need to make only few assumptions about landscape topography and about the DNA sequences that each landscape contains. The latter is particularly important because the set of sequences that a landscape contains determines the types of mutations that can occur along a mutational path to an adaptive peak. That is, landscapes can exhibit a composition bias—a statistical enrichment of a particular type of mutation relative to a null expectation, throughout an entire landscape or along particular mutational paths —that is independent of any bias in the mutation process. Our results reveal the way in which composition bias interacts with biases in the mutation process under different population genetic conditions, and how such interaction impacts fundamental properties of adaptive evolution, such as its predictability, as well as the evolution of genetic diversity and mutational robustness.

## Author summary

Mutation is often depicted as a random process due its unpredictable nature. However, such randomness does not imply uniformly distributed outcomes, because some DNA sequence changes happen more frequently than others. Mutation bias can be an orienting factor in adaptive evolution, influencing the mutational trajectories populations follow toward higher-fitness genotypes. Because these trajectories are typically just a small subset of all possible mutational trajectories, they can exhibit composition bias—an enrichment of a particular kind of DNA sequence change, such as transition or transversion mutations. Here, we use empirical data from eukaryotic transcriptional regulation to study how mutation bias and composition bias interact to influence adaptive evolution.

**Data Availability Statement:** The data are publicly available at http://thebrain.bwh.harvard.edu/uniprobe/academic-license.php and http://cisbp.ccbr.utoronto.ca/index.php, see Methods for details. Additionally, Genotype-phenotype

landscape files and commented code are publicly available at https://github.com/alejvcano/PLoSCompBio2020.

**Funding:** J.L.P. acknowledges support from Swiss National Science Foundation (Grant No. PP00P3_170604) http://www.snf.ch/en/Pages/default.aspx. The funders had no role in study design, data collection and analysis, decision to publish, or preparation of the manuscript.

**Competing interests:** The authors have declared that no competing interests exist.

# Introduction

Mutation exhibits many forms of bias, both in genomic location and toward particular DNA sequence changes [1]. For instance, a bias toward transitions (mutations that change a purine to a purine, or a pyrimidine to a pyrimidine), relative to transversions (mutations that change a purine to a pyrimidine, or vice versa; Fig 1a), has been widely observed in studies of mutation spectra, such as those based on reversion assays [2, 3], mutation accumulation experiments [4–6], sequence comparisons of closely related species [7–10], and analyses of putatively neutral polymorphisms in natural populations [11]. Because mutation provides the raw material of evolution, mutation bias may influence adaptive evolutionary change [12–14]. Indeed, transition bias has influenced the adaptive evolution of phenotypes as different as antibiotic resistance in *Mycobacterium tuberculosis* [15] and increased hemoglobin-oxygen affinity in high-altitude birds [16].

Adaptive evolution is often conceptualized as a hill-climbing process in a genotype-phenotype landscape, in which each location or coordinate corresponds to a genotype in an abstract genotype space, and the elevation of each location corresponds to fitness or some related quantitative phenotype [17, 18]. The topography of a genotype-phenotype landscape influences a wide range of evolutionary phenomena [19–21], including the evolution of genetic diversity, mutational robustness, and evolvability, as well as the predictability of the evolutionary process itself [19]. It also influences landscape navigability—the ability of an evolving population to reach the global adaptive peak via DNA mutation and natural selection [22]. Smooth, single-peaked landscapes are highly navigable, whereas rugged landscapes are not, because evolving populations can become trapped on local peaks [23], which frustrates further adaptive change [19, 21, 24].

Previous work has studied the interplay of mutation bias and landscape topography, and its influence on adaptive evolution, using synthetic genotype-phenotype landscapes [25, 26]. These studies have revealed that in single-peaked landscapes, mutation bias does not influence navigability, although it can influence the adaptive trajectory to the global peak, such that the average composition of the sequences in the trajectory reflects the sequence bias of the mutation process [26]. In this sense, mutation bias can be thought of as an "orienting factor" in evolution [25], which may affect predictability by making some mutational trajectories more likely than others. For example, in a simple two-locus model with two adaptive peaks of different heights, mutation bias increases the probability with which an evolving population converges on the suboptimal, but mutationally-favored peak [25]. Mutation bias is therefore capable of influencing the navigability of rugged landscapes. The extent to which it does, however, depends upon population genetic conditions [25]. Specifically, when the mutation supply is low, the first adaptive mutation to reach a substantial frequency is likely to go to fixation ("first-come-first-served"), and a bias in mutation supply will therefore influence which genetic changes drive adaptation. In contrast, when the mutation supply is high, the fittest adaptive mutation is likely to go to fixation ("pick-the-winner"), and a bias in mutation supply will have less of an effect on adaptation.

The study of genotype-phenotype landscapes is currently being transformed by methodological advances, including those in genome editing and in massively parallel assays such as deep mutational scanning [15, 20, 24, 27]. These facilitate the assignment of phenotypes to a large number of genotypes, and thus allow for the construction of empirical genotype-phenotype landscapes directly from experimental data. Examples include the "splicing-in" of exons [28], the binding preferences and enzymatic activities of macromolecules [29–32], the gene expression patterns of regulatory circuits [33], and the carbon utilization profiles of metabolic pathways [34]. In nearly all of these examples, the genotype-phenotype landscape is necessarily

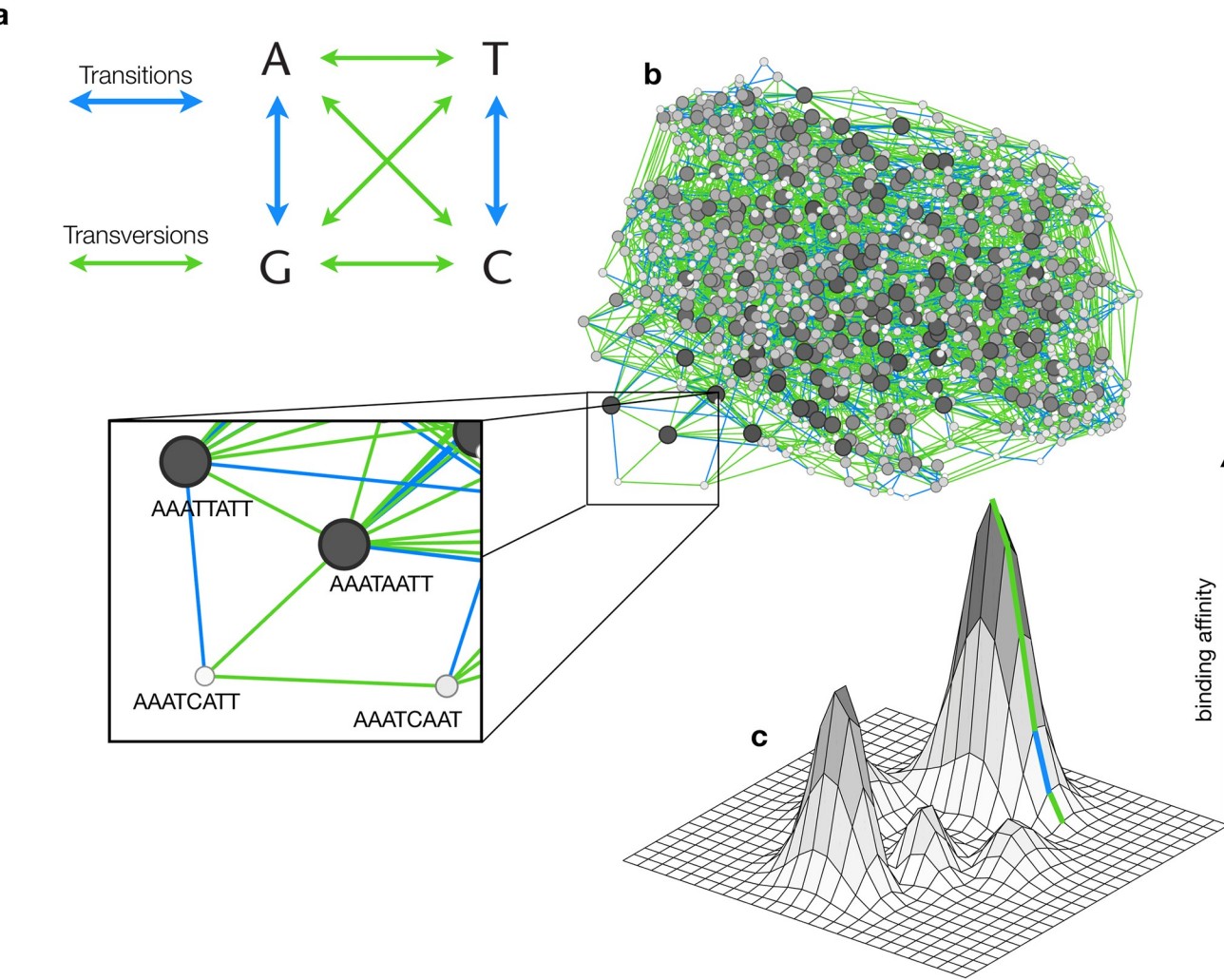

**Fig 1. Genotype-phenotype landscapes of transcription factor binding affinities and their composition bias. (a)** Transitions are mutations from a purine to a purine, or from a pyrimidine to a pyrimidine. Transversions are mutations from a purine to a pyrimidine, or vice versa. **(b)** The dominant genotype network for the yeast transcription factor Sum1. Each node corresponds to a DNA sequence that binds Sum1 with an E-score >0.35. Node size is proportional to number of connections (bigger = more) and color to binding affinity (darker = higher). Two nodes are connected by an edge if their corresponding sequences differ by a single point mutation (e.g., see inset), either a transition (blue edges) or a transversion (green edges). **(c)** Schematic representation of a genotype-phenotype landscape, and an accessible mutational path to the global peak involving four transversions and one transition.

incomplete, representing only a small fraction of a much larger landscape, which cannot be constructed in its entirety due to the hyper-astronomical size of the corresponding genotype space [18]. Complete landscapes—those in which a phenotype is assigned to all possible genotypes—can only be constructed for systems with sufficiently small genotype spaces.

Transcription factor-DNA interactions are one such system [22, 35–37]. Transcription factors are regulatory proteins that bind DNA to induce or inhibit gene expression [38]. The DNA sequences they bind are typically short (6-12nt) [39], which makes it possible to exhaustively characterize transcription factor binding preferences and thus to construct complete genotype-phenotype landscapes of transcription factor binding affinities [22]. In such landscapes, genotypes are short DNA sequences (transcription factor binding sites), and the phenotype of a sequence is its relative binding affinity for a transcription factor. Understanding how DNA mutations affect transcription factor binding affinity is important because such

mutations are commonly implicated in disease [40, 41], as well as in evolutionary adaptations and innovations [42, 43]. Previous work on these landscapes has revealed that they are highly navigable [22]. They tend to contain few adaptive peaks, which comprise binding sites that are both mutationally robust and accessible, meaning that it is typically possible to reach these peaks via a series of mutations that only move "uphill." These peaks often comprise multiple sequences, which facilitates the evolution of genetic diversity in high-affinity binding sites [22, 36]. Additionally, these landscapes often interface and overlap with one another, which has implications for the evolvability of transcription factor binding sites [35, 36].

There are two features that differentiate these landscapes from other empirical landscapes. First, they are complete. They comprise a measure of relative binding affinity for all possible DNA sequences of length eight. Second, data are available for many such landscapes, which facilitates statistical analyses of how landscape properties influence adaptive evolution. More-over, in contrast to synthetic landscapes, these empirical landscapes make very few assumptions about topography and about the DNA sequences each landscape contains. The latter is particularly important in the context of mutation bias, because the set of sequences a landscape contains determines the kinds of mutations that are present in adaptive mutational trajectories. For example, the TATA-binding protein binds sequences enriched for thymines and adenines, which means that most of the mutations present in this protein's genotype-phenotype landscape are transversions (A > T or T > A). Such composition bias—an enrichment of a particular type of mutation relative to a null expectation, throughout an entire landscape or along particular mutational paths—likely interacts with mutation bias to influence various aspects of adaptive evolution, such as landscape navigability, enhancing it when mutation is biased toward transversions, and hindering it when mutation is biased toward transitions. However, to our knowledge, the interaction between mutation bias and composition bias, and its influence on adaptive evolution, has not been studied, neither in the context of synthetic nor empirical genotype-phenotype landscapes.

Here, we study this interaction and its influence on adaptive evolution using 746 empirical genotype-phenotype landscapes of transcription factor binding affinities, under the assumption of selection for increased binding affinity (although selection does not always act to increase binding affinity [44, 45]). We find that when the mutation supply is low, mutation bias can increase or decrease landscape navigability as well as the predictability of evolution, depending upon whether mutation bias is aligned with composition bias in the adaptive trajectories to the global peak (i.e., both forms of bias are toward the same type of mutation—transitions or transversions). When the mutation supply is high, mutation bias does not influence navigability, as expected on theoretical grounds [25], but it can influence how a population is distributed throughout the landscape, which has implications for the evolution of genetic diversity, mutational robustness, and evolvability. Taken together, our results show that mutation bias and composition bias interact to influence adaptive evolution under a broad range of population genetic conditions.

## Results

### Genotype-phenotype landscapes exhibit composition bias in accessible mutational paths

We used protein-binding microarray data [46, 47] to construct empirical genotype-phenotype landscapes of transcription factor binding affinities for 746 transcription factors from 129 eukaryotic species, representing 48 distinct DNA binding domain structural classes (Methods; S1 Table). For each transcription factor, these data include an enrichment score (*E*-score)—a proxy for relative binding affinity—for all possible 32, 896 DNA sequences of length eight. We

constructed one landscape per transcription factor, using only those DNA sequences that specifically bound the transcription factor, as indicated by an *E*-score exceeding 0.35 [22, 35, 36]. We represented each landscape as a genotype network, in which nodes are transcription factor binding sites and edges connect nodes if their corresponding sequences differ by a single point mutation (Fig 1b) [35]. For some transcription factors ($\sim$ 37%), the genotype network fragmented into several disconnected components. When this occurred, we only considered the largest component, which always comprised more than 100 bound sequences. We refer to this as the dominant genotype network. We discarded non-dominant components because they usually comprised few sequences and rarely met our size requirement of 100 sequences. (S1 Fig). Each dominant genotype network formed the substrate of a genotype-phenotype landscape, whose surface was defined by the relative affinities (*E*-scores) of the network's constituent binding sites [22]. We accounted for noise in the protein binding microarray data using a noise threshold $\delta$, which allowed us to determine whether two DNA sequences differed in their binding affinity [22] (Methods). We refer to a mutation as accessible if it increases binding affinity more than $\delta$, and we refer to a series of accessible mutations as an accessible mutational path (Fig 1c).

We developed a measure of composition bias on the unit interval, which we applied to entire landscapes and to accessible mutational paths (Methods). It is based on the null expectation that one transition occurs per every two transversions. When this measure equals 0.5, there is no composition bias. Values below 0.5 imply a composition bias toward transversions, whereas values above 0.5 imply a composition bias toward transitions. Fig 2a shows the distribution of composition bias across all 746 landscapes. When considering all mutations in a landscape, we observed variation in composition bias ranging from 0.19 to 0.75, but as a whole, this distribution did not differ from our null expectation of one transition per every two transversions (black distribution in Fig 2a, one-sample *t* test, $t_{745} = -0.4310$, $p = 0.66$). In

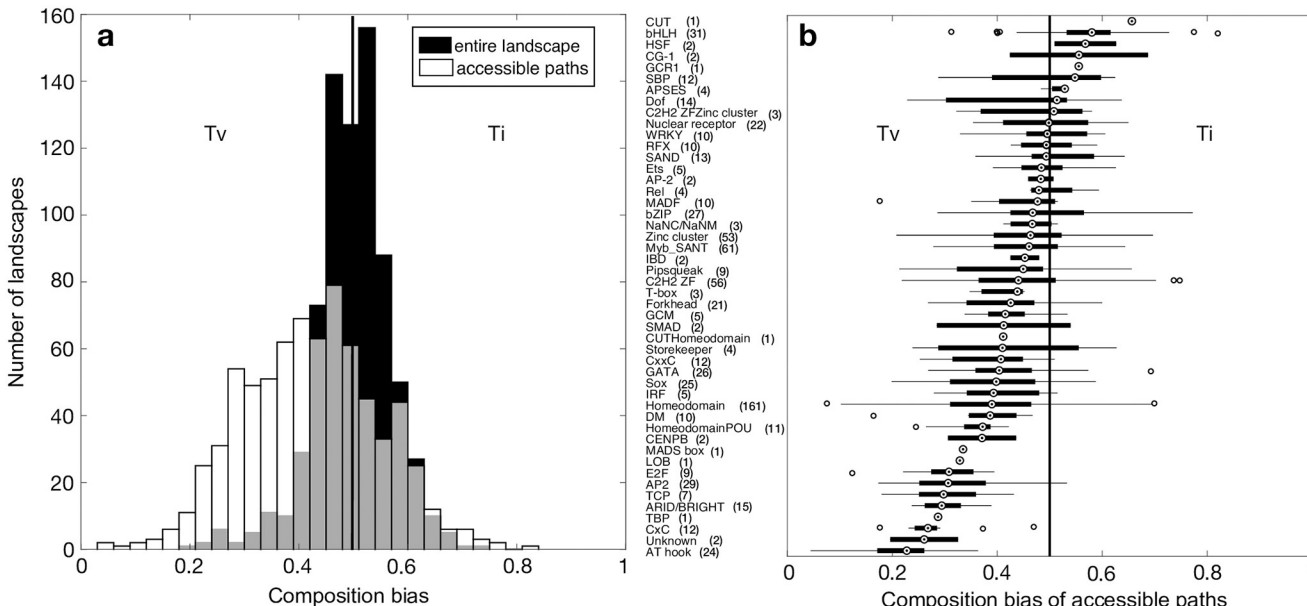

**Fig 2. Accessible mutational paths exhibit composition bias toward transversions. (a)** The black bars show the distribution of composition bias across entire landscapes. The white bars show the distribution of composition bias across accessible mutational paths to the global peak, starting from the 10% of binding sites with the lowest affinities in each landscape. Gray indicates the overlap in the distributions. Data pertain to all 746 landscapes. **(b)** Composition bias of accessible mutational paths, with landscapes grouped by DNA binding domain structural class. Numbers in parentheses indicate the number of transcription factors per class in our dataset.

contrast, when we considered accessible mutational paths to the global peak, we found significant composition bias toward transversions (white distribution in Fig 2a, one-sample $t$ test, $t_{745} = -14.3941$, $p < 10^{-40}$). This bias was sometimes extreme. For example, for 68 transcription factors, fewer than 1 in 7 mutations on an accessible mutational path were transitions, more than a three-fold decrease relative to our null expectation. Such composition bias varied among transcription factors from different DNA binding domain structural classes (Fig 2b). For example, AT hook transcription factors had a strong bias toward transversions, because their binding sites are enriched for adenine and thymine bases, and thus have a very low GC content. At the opposite extreme were transcription factors with a basic helix-loop-helix domain (bHLH), which had a bias toward transitions, because the sequences they bind are more neutral in terms of GC content (average GC content = 0.61). Landscapes with either many high or many low GC content sequences were more likely to exhibit composition bias toward transversion mutations (S2 Fig). Overall, we observed more extreme bias toward transversions than toward transitions in accessible mutational paths (Fig 2a), a bias that became even more pronounced as we increased the noise parameter $\delta$ (S3 Fig). This is because transversions tend to cause larger changes in binding affinity [48], and thus have larger regulatory effects [49]—a phenomenon we observed mainly near the global peak (S4 Fig). Because these distributions deviate so strongly from our null expectation, we focus on the composition bias of accessible mutational paths in all subsequent analyses, and we refer to this simply as composition bias for brevity.

To test the sensitivity of these observations to our use of $E$-score as a proxy for relative binding affinity, we considered an alternative proxy—the median signal intensity $Z$-score—which is available for 713 of the 746 transcription factors in our dataset. We found a strong correlation between the composition bias calculated from the two scores (Pearson's correlation coefficient $r = 0.7212$, $p < 10^{-10}$) (S5 Fig), and although composition bias often varied quantitatively among the two scores, it did not often vary qualitatively. That is, it was uncommon for landscapes to switch from exhibiting a composition bias toward transversions to a composition bias toward transitions, or vice versa. There were only 83 transcription factors (12%) for which such switching occurred, and of these roughly half ($\sim 48\%$) exhibited little to no composition bias when using $E$-scores (in the range (0.45, 0.55)). We therefore used $E$-scores for the rest of our analyses. In addition, we tested the sensitivity of our results to an alternative, less conservative definition of an accessible mutational path, which included any mutational steps that did not decrease binding affinity beyond some threshold ($\delta$—our noise parameter). S6 Fig shows that this alternative definition of accessibility results in composition biases that are highly similar to those observed under our initial more conservative definition, which we therefore use for the rest of our analyses, unless otherwise noted.

## Mutation bias interacts with composition bias to influence landscape navigability

We first explored how mutation bias and composition bias interact to influence adaptive evolution when the mutation supply is low and selection is strong. Under these population genetic conditions, only one mutation is present in the population at any time [50], which makes the process amenable to modeling as a random walk in genotype space (Methods). In this framework, each time step corresponds to the number of generations needed for a mutation to go to fixation, and the fixation probability of a mutant is proportional to its binding affinity and to the likelihood of that particular type of mutation occurring. The latter was determined by a mutation bias parameter $\alpha$, which is defined similarly to our measure of composition bias (Methods). Specifically, when $\alpha = 0.5$ there is no bias in mutation supply, values of $\alpha$ below 0.5

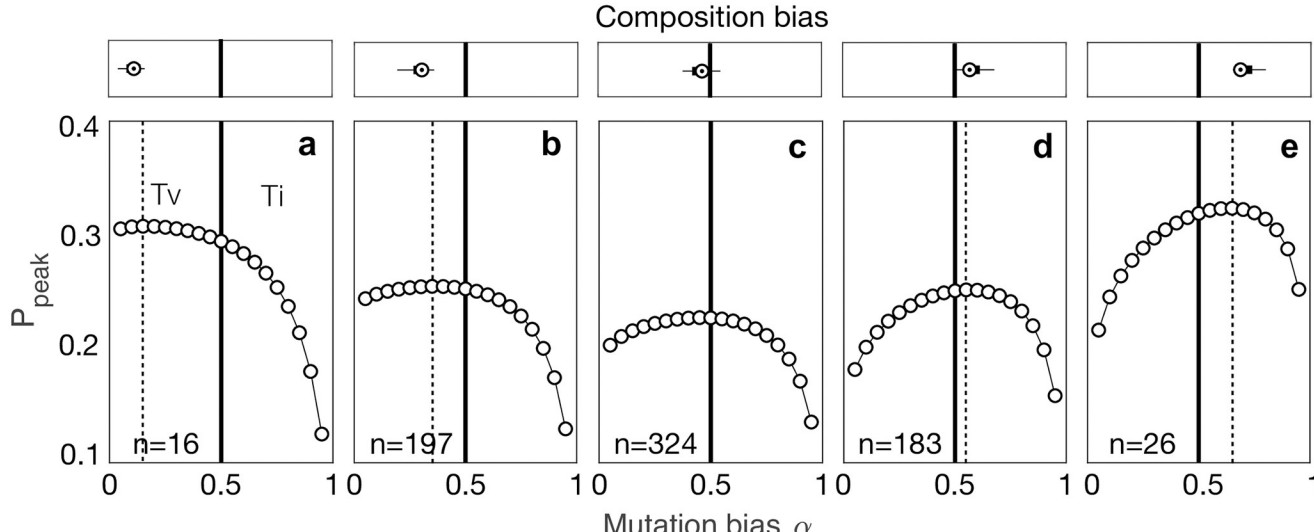

**Fig 3. Mutation bias interacts with composition bias to influence landscape navigability.** The probability $P_{peak}$ of reaching the global peak is shown for 19 different values of the mutation bias parameter $\alpha$. The solid vertical lines indicate no bias in mutation supply ($\alpha = 0.5$) and the dashed vertical lines indicate the value of $\alpha$ that maximizes $P_{peak}$. Landscapes are grouped based on their composition bias and the distribution of composition bias per panel is shown on top of each panel. The number of landscapes per panel is indicated is the bottom left corner.

mean that the mutation supply is biased toward transversions, whereas values above 0.5 mean that the mutation supply is biased toward transitions.

As a measure of landscape navigability, we calculated the probability of reaching the global peak (Methods). Fig 3 shows how mutation bias and composition bias interact to influence landscape navigability. We found that mutation bias could either enhance or diminish landscape navigability, relative to an unbiased mutation supply, depending upon whether the bias in mutation supply aligned with the composition bias of the accessible mutational paths in the landscape (i.e., the two forms of bias were toward the same kind of mutations—transitions or transversions). This effect is clearly seen when comparing the value of mutation bias that maximizes the probability of reaching the global peak across the five panels of Fig 3 (dashed vertical lines), which group landscapes according to their composition bias. S7 Fig shows the correlation between the mutation bias that maximizes $P_{peak}$ and composition bias, measured using two different definitions of accessible mutational paths. Landscapes with extreme composition bias exhibited increased sensitivity to mutation bias, relative to landscapes without composition bias. For example, for landscapes with a strong composition bias toward transversions (Fig 3a), the probability of reaching the global peak increased from 0.12 with a strong mutation bias toward transitions to 0.31 with a strong mutation bias toward transversions—a 2.6-fold increase. In contrast, for landscapes without composition bias (Fig 3c), only a 1.7-fold increase in the probability of reaching the global peak was obtained by varying the bias in mutation supply.

Overall, landscapes with strong composition bias were more navigable than those with intermediate or no bias, in that they had higher probabilities of reaching the global peak. We reasoned this is because landscapes with strong composition bias tend to contain fewer binding sites than those with intermediate or no bias (S8 Fig), and the probability of evolving to a landscape's global peak decreases with the number of binding sites in the landscape (as shown in S9 Fig for an unbiased mutation supply). In addition, our finding that navigability will be enhanced when mutation bias and composition bias are aligned and diminished otherwise is

insensitive to whether selection acts to increase or decrease binding affinity (S10 Fig). This is important because low-affinity binding sites often drive gene expression, for example in developmental enhancers [44, 45].

## Mutation bias interacts with composition bias to influence the predictability of evolution

When the mutation supply is low, a bias in mutation supply can influence an evolving population's adaptive trajectory through a genotype-phenotype landscape, making some mutational paths more likely than others [25, 26]. Mutation bias may therefore influence the predictability of evolution. We next explored how mutation bias interacts with composition bias to influence the predictability of evolution, quantifying predictability using a measure of path entropy [51] (Methods). This measure takes on low values when an evolving population tends to take few mutational paths to the global peak, each with high probability. It takes on high values when an evolving population tends to take many mutational paths to the global peak, each with low probability. It is therefore inversely related to the predictability of evolution. S11 Fig shows the relationship between path entropy and the mutation bias parameter $\alpha$ for 50 randomly chosen landscapes.

As expected, we found that mutation bias is considerably more likely to increase the predictability of evolution than to decrease it, in line with the notion of mutation bias as an orienting factor in evolution [25]. Across all landscapes, there was at least one mutation bias value that decreased path entropy relative to when there was no mutation bias (Fig 4a), and on average 73% of mutation bias values decreased path entropy relative to when there was no mutation bias (S12 Fig). Moreover, a mutation bias toward transitions minimized path entropy more often than one toward transversions (Fig 4a). Mutation bias therefore readily increases the predictability of evolution. Fig 4b shows how the misalignment between mutation bias and composition bias usually leads to the minimization of path entropy, thus increasing predictability. For landscapes with little to no composition bias, a strong mutation bias toward transitions was more likely to minimize path entropy than a strong mutation bias toward transversions. This is because in landscapes with no composition bias there are twice as many transversions than transitions, so a mutation bias toward transitions represents a greater evolutionary constraint in such cases. To determine the extent to which entropy changes in response to mutation bias, we calculated the ratio of the entropy observed in the absence of mutation bias to the minimum entropy caused by mutation bias (entropy$_{no\ bias}$/entropy$_{min}$), for each landscape. Path entropy decreased by an average of approximately 2-fold and 3-fold in the most extreme mutation biases toward transversions and transitions, respectively (Fig 4c).

Perhaps counter-intuitively, we found that mutation bias can also decrease the predictability of evolution. For 693 landscapes (93%), there was at least one mutation bias value that increased path entropy relative to when there was no mutation bias (Fig 4d), and on average across all 746 landscapes, 27% of mutation bias values increased path entropy relative to when there was no mutation bias (S12 Fig). To explore this further, we determined the mutation bias parameter $\alpha$ that maximized path entropy for each landscape. Fig 4d shows the distribution of this parameter, which varied across the full range of $\alpha$ values considered. We hypothesized that the value of $\alpha$ that maximizes path entropy will be positively correlated with composition bias, such that path entropy will be maximized when these biases are aligned. Fig 4e reveals this positive correlation, which is statistically significant, but weak in explanatory power (Spearman's rank correlation coefficient $\rho = 0.13$, $p < 10^{-4}$). The reason is that our measure of composition bias does not capture how mutations are distributed across accessible mutational

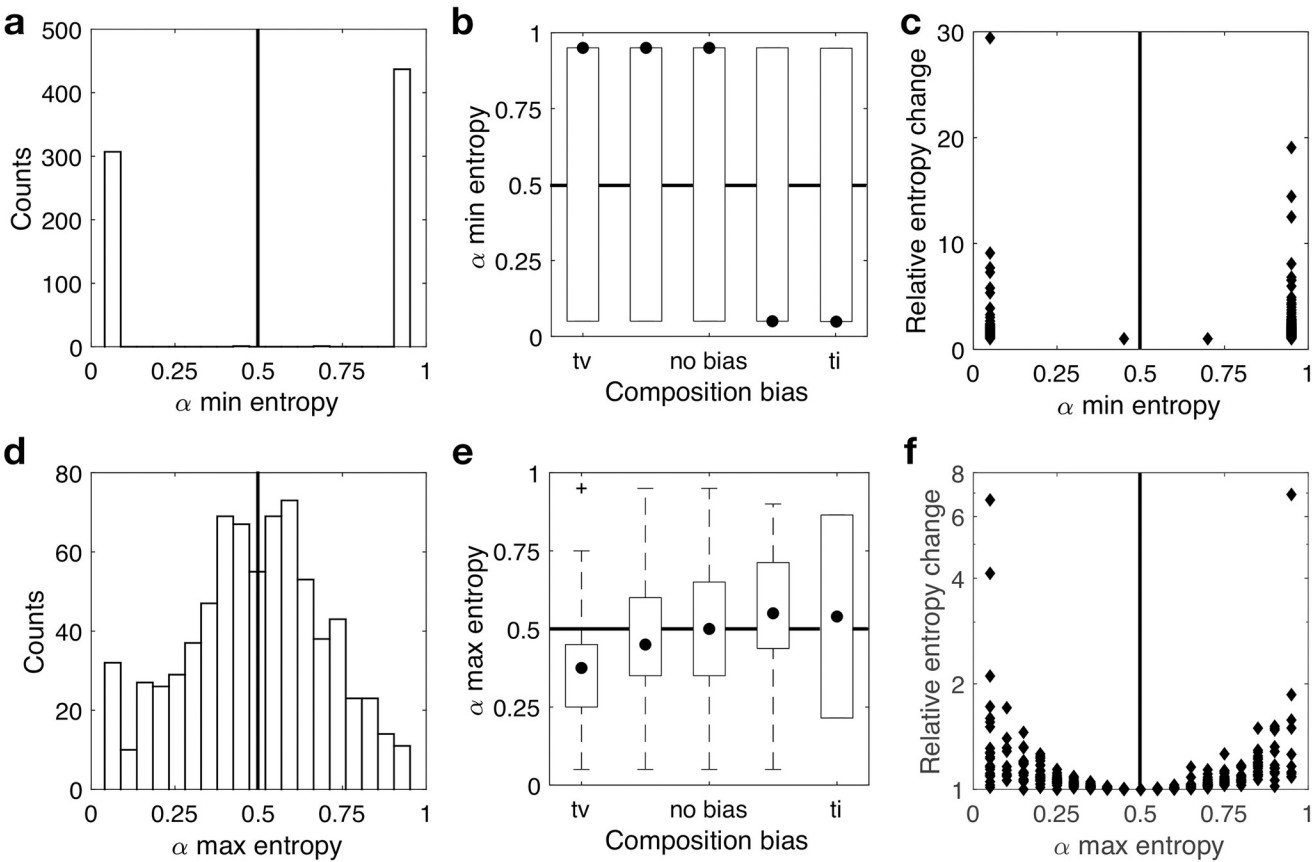

**Fig 4. Mutation bias interacts with composition bias to impact the predictability of evolution. (a)** Distribution of the mutation bias parameter $\alpha$ that minimizes path entropy for each landscape. **(b)** Mutation bias parameter that minimizes path entropy, shown in relation to composition bias. **(c)** Relative entropy change (entropy$_{no\ bias}$/entropy$_{min}$), shown in relation to the mutation bias parameter $\alpha$ that minimizes path entropy. **(d)** Distribution of the mutation bias parameter $\alpha$ that maximizes path entropy for each landscape. **(e)** Mutation bias parameter that maximizes path entropy, shown in relation to composition bias. **(f)** Relative entropy change (entropy$_{max}$/entropy$_{no\ bias}$), shown in relation to the mutation bias parameter $\alpha$ that maximizes path entropy. Data in all panels pertain to all 746 landscapes.

paths, which strongly influences the value of $\alpha$ that maximizes path entropy (S13 Fig). In addition, Fig 4f shows the relative change in entropy (entropy$_{max}$/entropy$_{no\ bias}$) in relation to the mutation bias parameter $\alpha$ that maximized path entropy. It reveals that path entropy increased by up to 7-fold, and that the largest increases were associated with the most extreme biases in mutation supply, either toward transitions or toward transversions. In these cases, evolution became less predictable because an evolving population could traverse a greater diversity of accessible mutational paths. In sum, these analyses reveal that mutation bias and composition bias interact to influence the predictability of evolution, in most cases increasing predictability, but in many others decreasing it. Thus, whether mutation bias acts as an orienting or dispersive factor in evolution depends upon the prevalence and type of composition bias in the landscape.

## Mutation bias influences the distribution of polymorphic populations in genotype-phenotype landscapes

We next explored how mutation bias influences adaptive evolution when the mutation supply is high and selection is strong. Under these population genetic conditions, multiple mutations

coexist in the population and compete for fixation. Because this process is challenging to model analytically, we used computer simulations of a Wright-Fisher model of evolutionary dynamics (Methods). As in our previous analyses, we used the probability of reaching the global peak as a measure of landscape navigability. We found that mutation bias has no effect on navigability in this "pick-the-winner" regime (S14 Fig), as expected on theoretical grounds [25]. However, we reasoned that mutation bias may still influence other properties of an evolving population, specifically those that depend upon the population's distribution in the landscape. To explore this possibility, we used a measure called an overlap coefficient, which quantifies the similarity of two populations as the proportion of individuals that are common to both (Methods). This coefficient takes on its minimum value of 0 when there are no individuals in common between two populations; it takes on its maximum value of 1 when both populations are identical, having the same individuals in the same proportions. We applied this measure to pairs of populations after they had evolved for 1000 generations, reaching steady state (S15 Fig). As a baseline for comparison, we first calculated the overlap coefficient for pairs of replicate populations. That is, pairs of populations with identical initial conditions, but with different random number generator seeds (Fig 5a). This allowed us to assess how different we expect two evolved populations to be at steady state, due solely to the stochastic nature of the evolutionary simulations. For replicate populations, the overlap coefficient ranged from 0.912 to 1, with a median of 0.976 and a 75th percentile of 0.817 (Fig 5b). This indicates that while the stochastic nature of the evolutionary simulation can cause large changes in a polymorphic population's distribution in a landscape, it usually does not. Replicate populations tend to converge on highly similar distributions. In contrast, when we calculated the overlap coefficient for pairs of evolved populations with identical initial conditions (including identical random number generator seeds), but different values of the mutation bias parameter, we observed far less overlap (two-sample Kolmogorov-Smirnov test, $D = 0.2695$, $p < 10^{-31}$). Specifically, the overlap coefficient ranged from 0.692 to 1, with a median of 0.908 and a 75th percentile of 0.274 (Fig 5b). This indicates that mutation bias often has a strong influence on an evolved population's distribution in a genotype-phenotype landscape. The strength of this influence depends upon how different the mutation bias values are in the populations being compared (Fig 5c), with larger differences corresponding to larger changes in population distribution, a trend that also holds for infinitely large populations (S16 Fig).

## Mutation bias and composition bias interact to influence the evolution of genetic diversity and mutational robustness in polymorphic populations

We next asked how mutation bias interacts with composition bias to influence the evolution of genetic diversity. We reasoned that when mutation bias is aligned with composition bias, evolving populations will be less constrained in their exploration of the landscape and will therefore accumulate greater genetic diversity. To explore this possibility, we measured the genetic diversity of populations at steady state using Shannon's diversity index (Methods). This measure takes on its maximum value of 1 when the population comprises all possible individuals in equal proportions. For DNA sequences, this means that all four bases are equally likely to appear at all positions in the sequence. The measure takes on its minimum value of 0 when the population comprises $N$ copies of just a single individual. Fig 6a–6e shows that mutation bias can either increase or decrease genetic diversity, relative to an unbiased mutation supply, and depending on whether mutation bias aligns with composition bias. This effect was most pronounced in landscapes with strong composition bias, either toward transversions (Fig 6a) or transitions (Fig 6e), and when the mutation supply was high. For example, in landscapes with a strong composition bias toward transversions (Fig 6a), a bias in mutation supply could

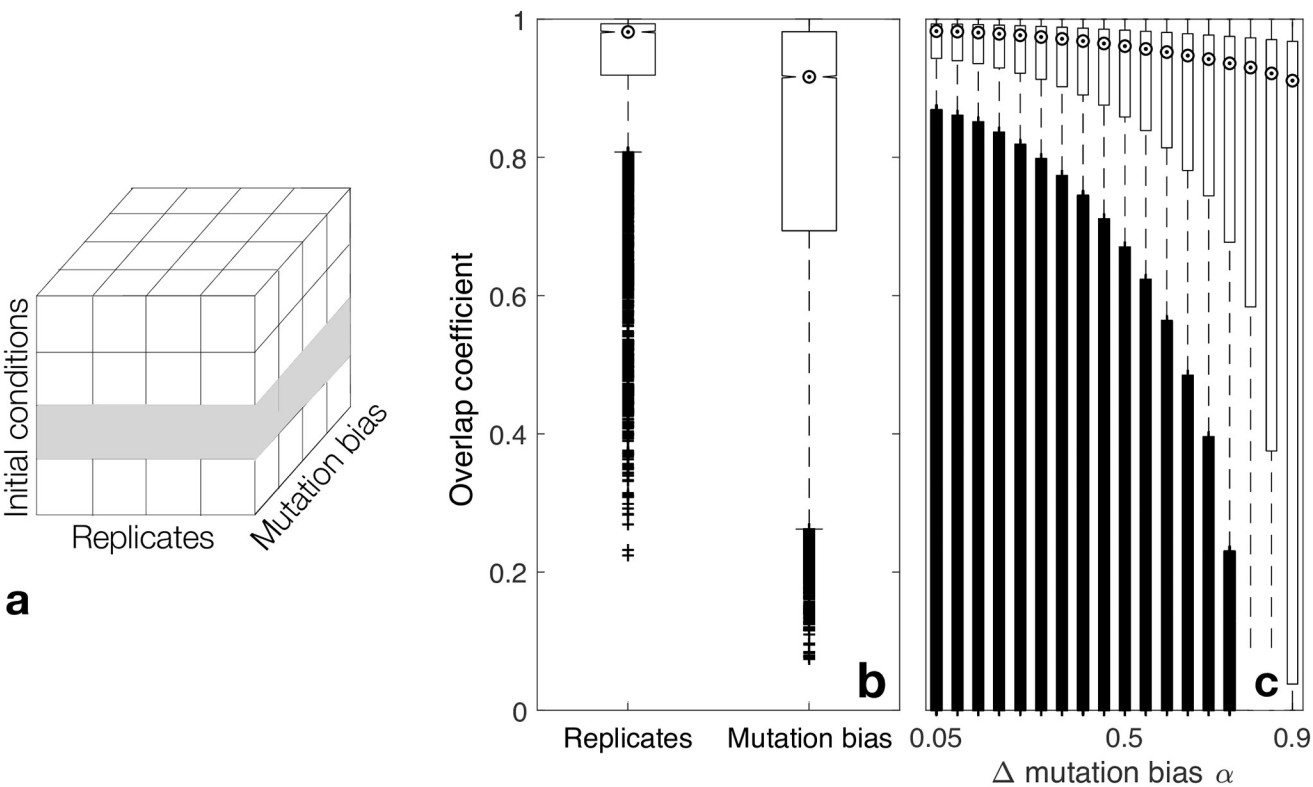

**Fig 5. Evolving polymorphic populations are more sensitive to changes in mutation bias than to the stochastic nature of the evolutionary simulations. (a)** Schematic figure of our experimental design. For each landscape and combination of population size and mutation rate ($N\mu$), we considered 10 replicates for each of 10 different initial conditions and 19 values of the mutation bias parameter $\alpha$. Importantly, we used the replicate number to seed the random number generator of each evolutionary simulation, facilitating the comparison of variation across replicates versus across the mutation bias parameter $\alpha$. For example, the matrix elements indicated in gray contain the information necessary to compare the effects of the mutation bias parameter with the stochasticity of the evolutionary simulations, for one initial condition. **(b)** Overlap coefficient for pairs of evolved populations that differ in random number generator seed ("Replicates") or in mutation bias parameter ("Mutation bias"). Notches indicate medians, whiskers indicate the 25th and 75th percentiles, and cross symbols indicate outliers. **(c)** Overlap coefficient for pairs of evolved populations, shown in relation to the difference in their mutation bias parameters.

change genetic diversity 1.9-fold when $N\mu = 50$, but had almost no effect when $N\mu = 5$. In these cases, genetic diversity could reach levels higher than those observed on landscapes with little to no bias (Fig 6c). Calculating genetic diversity using nucleotide diversity $\pi$ results in similar patterns (S17 Fig). Specifically, diversity increases when mutation bias aligns with composition bias and decreases otherwise. To further illustrate how mutation bias and composition bias interact to influence population diversity, we show in S18 Fig the allele frequency spectra of populations evolved on two different landscapes, which we chose as illustrative examples because of their strong composition bias toward transversions (S18a Fig) and toward transitions (S18b Fig). As expected, the allele frequency spectra reflect the sequence bias of the mutation process, with more transition polymorphisms present when mutation is biased toward transitions and more transversion polymorphisms present when mutation is biased toward transversions. These examples also illustrate two different ways in which the interaction between mutation bias and composition bias influence the frequency of mutations in the population. In S18a Fig, polymorphisms are present, but they are rare, and the sequence that evolves to the highest frequency is the same as in the absence of mutation bias. In contrast, in S18b Fig some transition polymorphisms come to dominate the population, such that the

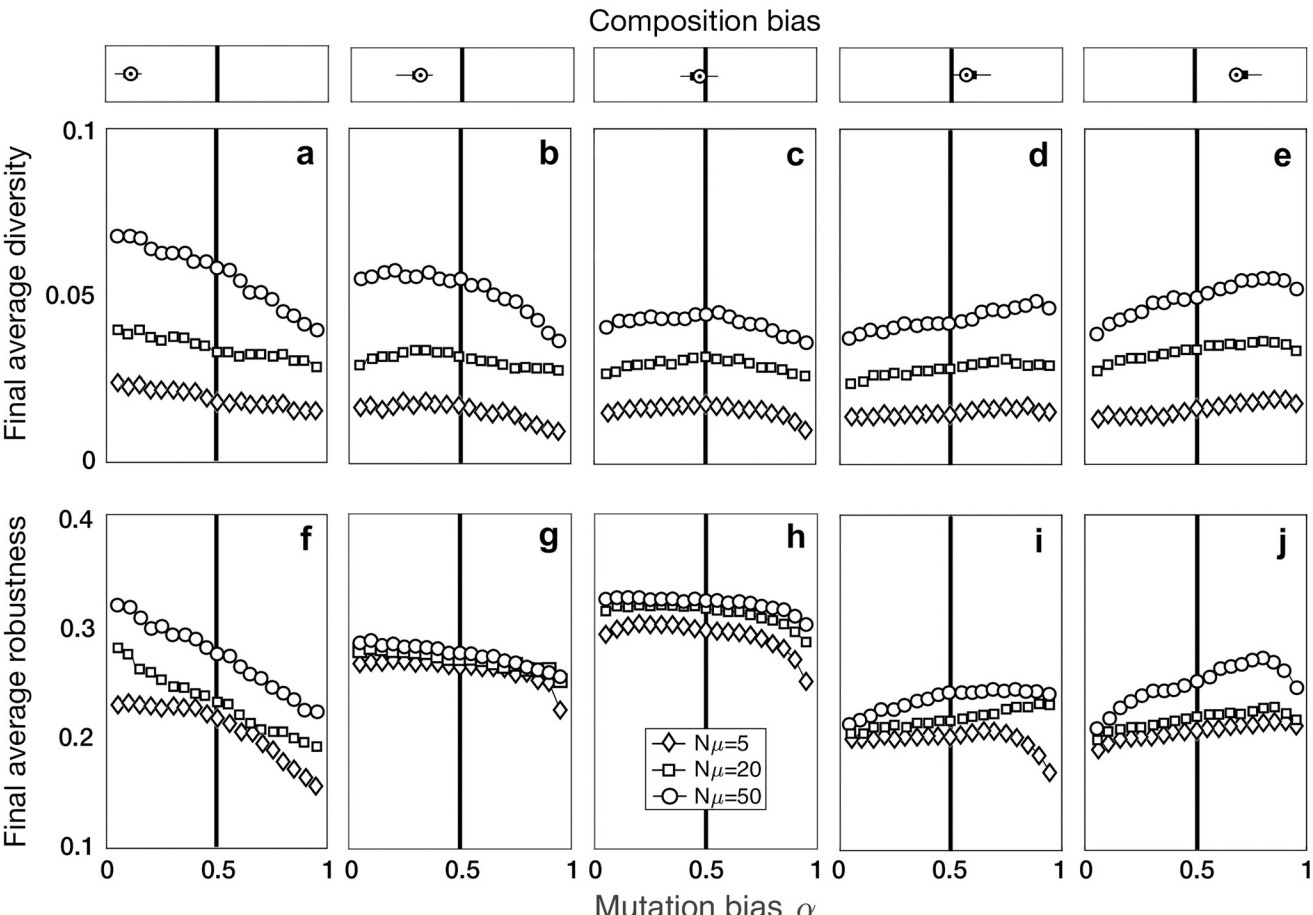

**Fig 6. Mutation bias interacts with composition bias to influence the evolution of genetic diversity and mutational robustness. (a-e)** The average genetic diversity and **(f-j)** mutational robustness of evolved populations at steady state is shown for 19 different values of the mutation bias parameter $\alpha$, and for each of three different values of mutation supply $N\mu$ (see legend). The solid vertical lines indicate no bias in mutation supply ($\alpha = 0.5$). Landscapes are grouped based on their composition bias and the distribution of composition bias per panel is shown on top of each panel as in Fig 3.

sequence that evolves to the highest frequency is not the same as in the absence of mutation bias.

We then explored the potential consequences of these changes in genetic diversity. First, we characterized how mutation bias and composition bias interact to influence the mutational robustness of binding sites in evolved populations at steady state. We quantified the mutational robustness of an individual binding site as the fraction of all possible mutations to that site that created another site that was also part of the landscape [35, 37]. The mutational robustness of a population of binding sites was then simply calculated as the average mutational robustness of its constituent sites. Fig 6f–6j shows that mutation bias can increase or decrease the mutational robustness of an evolved population, relative to an unbiased mutation supply, especially in landscapes with strong composition bias, either toward transversions (Fig 6f) or transitions (Fig 6j). In contrast, for landscapes with little to no composition bias (Fig 6h), only the most extreme values of mutation bias influenced mutational robustness. For landscapes with strong composition bias, the changes in mutational robustness caused by mutation bias mirrored the changes observed for genetic diversity (compare Fig 6a and 6e to Fig 6f anf 6j). In these landscapes, therefore, a decrease in genetic diversity was associated with a decrease in mutational robustness. Moreover, we observed that populations evolving on landscapes with composition

bias tended to be less mutationally robust at steady state. The reason is that landscapes with composition bias tended to comprise genotypes with fewer mutational neighbors, relative to landscapes without composition bias (S19 Fig). When mutation bias aligned with composition bias, those genotypes with more mutational neighbors were more likely to evolve, because the mutation bias oriented the population toward those genotypes.

## Mutation bias influences the evolvability of polymorphic populations

Because mutational robustness is a cause of evolvability [21, 52, 53], we explored whether mutation bias and composition bias interact to influence evolvability—defined in this context as the ability of mutation to bring forth new binding phenotypes (Methods) [35]. To test this, we calculated the average number of transcription factors that bind the one-mutant neighbors of any individual in the population at steady state, and computed the difference between these averages for populations with a strong mutation bias toward transversions ($\alpha = 0.05$), relative to an unbiased mutation supply ($\alpha = 0.5$), as well as toward transitions ($\alpha = 0.95$), relative to an unbiased mutation supply. S20 Fig shows how a bias in mutation supply affected the evolvability of polymorphic populations for 128 transcription factors from *Arabidopsis thaliana* and 128 transcription factors from *Mus musculus*, the two species with the most transcription factors in our dataset. In approximately 70% of the landscapes, mutation bias had no influence on evolvability. However, in the remaining 30% of landscapes, mutation bias could increase or decrease the number of transcription factors accessible via point mutation. In *A. thaliana*, this ranged from plus or minus 6 transcription factors ($\sim 5\%$), whereas in *M. musculus* this ranged from plus 12 to minus 7 transcription factors (between $\sim 5\%$ and $\sim 9\%$). These observations suggest that mutation bias is capable of orienting evolving populations both toward and away from more evolvable regions of genotype-phenotype landscapes. However, this effect is apparently independent of composition bias. This may seem counterintuitive at first glance, because the interaction between mutation bias and composition bias influences both genetic diversity and mutational robustness, two properties that facilitate evolvability [53]. However, the global peaks of the landscapes considered here are sufficiently narrow that any increase of diversity or robustness within them is not sufficient to cause a change in the number of phenotypes accessible via mutation. Said differently, the fraction of sequence space covered by these peaks is too small to facilitate mutational access to the landscapes of other transcription factors. We speculate that in landscapes with broader, mesa-like peaks, mutation bias and composition bias could interact to influence evolvability.

## Discussion

The mapping of genotype to phenotype influences how mutation brings forth phenotypic variation [54]. Biases in this map can therefore influence evolution [55]. For example, so-called phenotypic bias—that most phenotypes are realized by few genotypes, but a few phenotypes are realized by many genotypes—can cause a phenomenon known as the "arrival of the frequent", wherein phenotypes evolve not because they are the most fit, but rather because they are the most common [56, 57]. Moreover, phenotypes can exhibit a bias in their mutational connectivity to other phenotypes, such that phenotype-changing mutations are more likely to lead to common phenotypes than to uncommon phenotypes, thus reducing the ability of mutation to bring forth phenotypic variation [58].

   Here, we used empirical genotype-phenotype landscapes of transcription factor binding affinities to study a different form of bias, namely composition bias. This describes the prevalence of a particular kind of mutation in a landscape, or in a subset of mutational paths in a landscape, relative to a null expectation. The kind of mutation we considered was a transition

mutation, and we measured its prevalence relative to the null expectation that one transition occurs for every two transversions. We found that composition bias is common among accessible mutational paths to a landscape's global adaptive peak, and that for a large diversity of transcription factor families, this bias is toward transversions. We showed that such composition bias can interact with mutation bias to influence the navigability of genotype-phenotype landscapes and the predictability of evolution, as well as the evolution of genetic diversity and mutational robustness.

Such interaction was most pronounced in landscapes with strong composition bias, and when the bias in mutation supply was either aligned or in opposition with composition bias. Estimates of base-substitution mutation spectra are available for at least five of the species studied here [59]: *Homo sapiens*, *Drosophila melanogaster*, *Caenorhabditis elegans*, *Arabidopsis thaliana*, and *Saccharomyces cerevisiae*. Transforming the reported transition:transversion ratios to our mutation bias parameter $\alpha$ results in values that range from roughly the null expectation of one transition per two transversions ($\alpha = 0.47$ for *C. elegans*) to a strong bias toward transitions (0.83 in *Arabidopsis thaliana*—a 4.8-fold increase over the null expectation). Our results therefore suggest that biologically-realistic values of mutation bias can influence the evolution of transcription factor binding sites, specifically for transcription factors whose genotype-phenotype landscapes exhibit composition bias, either toward transitions or transversions.

Ideally, we could detect the influence of such an interaction on the evolution of transcription factor binding sites *in vivo*. Previous work used single nucleotide polymorphism and functional genomics data to show how the topology and topography of a genotype-phenotype landscape influences the evolution of binding site diversity in *A. thaliana* and *S. cerevisiae* [22, 36]. For example, transcription factors with broad global peaks were found to exhibit more diversity in their high-affinity binding sites than transcription factors with narrow global peaks [22]. It would be desirable to use such data to study how mutation bias and composition bias interact to influence the evolution of transcription factor binding sites *in vivo*. For example, one could ask whether the binding sites of transcription factors with a strong composition bias toward transversions exhibit less diversity in genomic regions prone to transition mutations than in other genomic regions, as our results suggest they would. However, such an analysis is complicated because one would need to identify functional binding sites for the same transcription factor in genomic regions that differ in mutation bias, but not in mutation rate. While some regulatory regions are more prone to transition mutations than others, such as CpG-rich promoters, which are susceptible to $C > T$ transitions due to the spontaneous deamination of 5-methylcytosines, these regions also exhibit elevated mutation rates [60].

It may be possible to overcome this challenge with experiments. For example, a single low-affinity binding site for an activating transcription factor could be used to seed two separate populations of binding sites, where the two populations differ in their mutation bias. This could be achieved by introducing mutations with error-prone PCR, using enzymes that differ in their mutation spectra, but have similar mutation rates. The mutated binding sites could then be cloned into plasmids upstream of a reporter gene, such as yellow fluorescence protein, transformed into bacterial cells, and exposed to selection for increased fluorescence using flow-activated cell sorting. This process of mutation and selection could then be repeated for several rounds, cloning the mutated binding sites back into the ancestral plasmid backbone and transforming the plasmids into fresh bacterial cells before each round of selection to ensure that increased fluorescence is driven by mutations in the binding site, rather than by mutations in the protein or elsewhere in the bacterial genome. By comparing replicate experiments for different binding site seeds and for transcription factors that vary in their composition bias, it may be possible to determine if and how mutation bias and composition bias

interact to influence the *in vivo* evolution of increased affinity in transcription factor binding sites.

Our analysis makes two key assumptions, the caveats of which are worth highlighting. The first is our assumption of selection for increased binding affinity. Low-affinity binding sites are also commonly employed to regulate gene expression, particularly for the auto-regulation of high-copy number transcription factors in bacteria [61] and during the development of multi-cellular organisms [44, 45, 62]. The second assumption is that of a linear relationship between the selective advantage conferred by a mutation and the change in binding affinity that the mutation causes. In reality, this relationship is likely non-linear, site-specific, and dependent upon local transcription factor concentrations. Relaxing these two assumptions will transform the topographies of the landscapes studied here, and will alter the composition biases of their accessible mutational paths. Such transformations are therefore likely to affect landscape navigability. However, we do not anticipate they will affect the way in which mutation bias and composition bias interact to influence landscape navigability. Regardless of the particular topographical properties of the landscape under investigation, navigability will be enhanced when mutation bias and composition bias are aligned, and diminished otherwise.

While we focused our study on transcription factor-DNA interactions, composition bias is likely to exist in other genotype-phenotype landscapes as well. For example, many RNA binding proteins target sequences enriched for guanine and uracil [63], and their binding affinity landscapes will therefore exhibit a composition bias toward transversions. Additionally, composition bias is not limited to genotype-phenotype landscapes of intermolecular interactions as studied here, but it may also be present in the landscapes of some macromolecules. Finally, other forms of composition bias and mutation bias may interact to influence adaptive evolution, including GC:AT bias and deletion bias. The mutation bias signatures of various cancers [64], which influence the *de novo* evolution of transcription factor binding sites [65], may also interact with composition bias to influence landscape navigability. As the scope and scale of genome editing and deep mutational scanning studies continues to expand, we will gain a better understanding of the prevalence of composition bias in empirical genotype-phenotype landscapes and its potential to interact with mutation bias in shaping adaptive mutational trajectories.

## Materials and methods

### Data

We constructed genotype-phenotype landscapes using data from protein binding microarrays, which we downloaded from the UniPROBE [46] and CIS-BP [47] databases. These data include proxies for the relative binding affinity of a transcription factor to all possible $(4^8 - 4^4)/2 + 4^4 = 32,896$ eight-nucleotide, double-stranded DNA sequences (transcription factor binding sites). These proxies include an enrichment score (*E*-score), which for each sequence is a function of the fluorescence intensities of a subset of probes that contain the sequence and a subset of probes that do not contain that sequence [66], and a *Z*-score, which for each sequence is the difference between the logarithm of the median fluorescence intensities of a subset of probes that contain the sequence and the logarithm of the median fluorescence intensities of all probes, reported in units of standard deviation. We considered a sequence as specifically bound by a transcription factor if its *E*-score exceeded 0.35, following previous work [22, 35, 37, 67]. We included a landscape in our dataset if its dominant genotype network comprised at least 100 bound sequences. According to these criteria, our dataset included 746 transcription factors, representing 129 eukaryotic species, and 48 DNA-binding domain structural classes. Details are provided in S1 Table.

## Constructing and analyzing genotype-phenotype landscapes

For each transcription factor, we used the Genonets Server [68] to construct a genotype-phenotype landscape from the set of sequences that specifically bound the factor (i.e., with an E-score > 0.35). We represented each such sequence as a node in a genotype network, and connected nodes with edges if their corresponding sequences differed in a single point mutation. Note that we did not consider indels in our definition of a mutation, like we did in our previous work [22, 35–37]. The reason is that we were interested in understanding the influence of a form of composition bias defined by point mutations (transitions vs. transversions), and we were concerned that the inclusion of indel-based edges would confound our analyses. For genotype networks that were fragmented into multiple components, we only considered the largest component, which we call the dominant genotype network.

Each dominant genotype network served as the substrate of a genotype-phenotype landscape, the surface of which was defined by relative binding affinity. For our main analyses, this was captured by the *E*-score, whereas for some of our sensitivity analyses, this was captured by the *Z*-score. We studied accessible mutational paths to the global peaks of these landscapes. An accessible mutational path comprises edges that each confer an increase in binding affinity greater than the noise threshold parameter $\delta$ [22]. For each transcription factor, we calculated $\delta$ as the residual standard error of a linear regression between the affinity values of all bound sequences from the two replicate protein binding microarrays [22]. Thus, each transcription factor had its own $\delta$, which reflects the noise in the replicated measurements for that particular transcription factor.

## Mutation bias and composition bias

We report both mutation bias and composition bias relative to the null expectation that one transition occurs for every two transversions. Letting $Ti$ and $Tv$ represent mutation rates of transitions and transversions, respectively, we define mutation bias as

$$\alpha = \frac{Ti}{Ti + Tv/2}. \tag{1}$$

A mutation bias of $\alpha = 0.5$ corresponds to the null expectation of one transition per two transversions. Values below 0.5 mean there are more transversions than expected under the null, whereas values above 0.5 mean there are more transitions than expected under the null.

Composition bias was measured in the same way, except with $Ti$ and $Tv$ representing the number of transitions and transversions in a landscape, or in an accessible mutational path to the global peak of a landscape.

## Origin-fixation model of evolutionary dynamics

We used an origin-fixation model to study evolutionary dynamics when the mutation supply is low ($N\mu << 1$). This was implemented using Markov chains, a memoryless process that gives the jumping probability from one genotype $i$ to another genotype $j$ using the matrix

$$P_{i,j} = \frac{\phi_{i,j} f_{i,j}}{\sum_{\forall k} \phi_{i,k} f_{i,k}} \tag{2}$$

where $f_{i,j}$ is the relative difference in binding affinity $b$

$$f_{i,j} = \begin{cases} b_j/b_i - 1 & \text{if} \quad b_j > b_i \\ 0 & \text{otherwise.} \end{cases} \tag{3}$$

and

$$\phi_{i,j} = \begin{cases} \alpha & \text{if edge } (i,j) \text{ is a transition} \\ (1-\alpha) & \text{if edge } (i,j) \text{ is a transversion.} \end{cases} \quad (4)$$

Then in general, the probability of going from any state to another state in a Markov chain given by the matrix $P$ (Eq (2)) after $t$ steps is

$$(P^t)_{i,j}. \quad (5)$$

## Wright-Fisher model of evolutionary dynamics

We carried out simulations of a Wright-Fisher model to study evolutionary dynamics when the mutation supply is high ($N\mu > 1$). Each simulation was initialized with a monomorphic population comprising $N$ copies of the same sequence, chosen from the bottom 10% of binding affinity values in the landscape. In each generation $t$, $N$ sequences were chosen from the population at generation $t - 1$ with replacement and with a probability that was linearly proportional to binding affinity. Mutations were introduced to these sequences at a rate $\mu$ per sequence per generation with a mutation bias $\alpha$. For each of the 746 landscapes, we performed 15 replicate simulations for each initial condition, using 19 linearly spaced mutation bias values between 0.05 and 0.95, and 3 mutation supply values ($N\mu \in \{5, 20, 50\}$). Each simulation ran for 1000 generations, which was sufficient to ensure that the population had reached steady state.

## Landscape navigability

As a measure of landscape navigability, we calculated the probability of reaching the global peak, starting from the 10% of sequences in the dominant genotype network with the lowest binding affinity. For low mutation supply, this was calculated as the average probability of going from the initial sequences to the global peak using Eq (5) after $t = 1000$ steps. For high mutation supply, this was calculated as the fraction of simulations per landscape in which at least 50% of the population reached the global peak.

## Path entropy

PathMAN (Path Matrix Algorithm for Networks), is a publicly available Python script that efficiently calculates path statistics of a given Markov process [51]. We employed PathMAN to calculate the Shannon's entropy of the path distribution, which accounts for the predictability of the process. Low entropy means that few paths with large probability dominate the process, while large entropy means that several low-probability paths contribute. We calculated the path entropy for 19 different values of the mutation bias parameter within the range [0.05,0.95], in order to find the mutation bias parameters $\alpha_{\min}$ and $\alpha_{\max}$ that minimize and maximize the path entropy for each landscape.

## Overlap coefficient

The overlap coefficient between two different polymorphic populations $A$ and $B$ was calculated as

$$O_{A,B} = \frac{|C|}{\min(|A|, |B|)}, \quad (6)$$

where *A* and *B* are multisets—sets that permit multiple instances of an element. The cardinality of such multisets is defined as

$$|A| = \sum_{x \in A} m_a(x) \quad \text{and} \quad |B| = \sum_{x \in B} m_B(x), \tag{7}$$

where the number of occurrences of the element *x* in the multiset is indicated by the multiplicity function $m(x)$.

Then *C* is the multiset defined as $C = A \cap B$, with multiplicity function

$$m_C(x) = \min(m_A(x), m_B(x)) \qquad \forall x \in A \cup B. \tag{8}$$

For example, if $A = \{1, 1, 2, 2, 2, 3\}$ and $B = \{1, 2, 2, 4\}$, then $C = \{1, 2, 2\}$ and the overlap coefficient is $O_{A,B} = 0.75$.

## Quasispecies dynamics of infinite populations

Since the matrix in Eq (2) is non-negative and connected, the Perron-Frobenius theorem for non-negative matrices applies [69]. Hence, the steady-state distribution of an infinite size population on a genotype network is determined by the eigenvector associated to the largest eigenvalue of the matrix in Eq (2). Per each landscape, the eigenvectors were computed numerically for 19 different values of mutation bias parameter $\alpha$ within the range [0.05,0.95].

## Genetic diversity

We measured the diversity of a population as the average Shannon's diversity over all genotypes, normalized by the maximum diversity per landscape $\log_2(n)$:

$$H = \frac{-\sum_i p_i \log_2 p_i}{\log_2(n)} \tag{9}$$

where *n* is the number of sequences in the landscape, $p_i$ is the fraction of the steady-state population that is at sequence *i*.

## Evolvability

We quantified the evolvability of a transcription factor's binding sites as follows. First, we determined the set of binding sites that have evolved at steady state in our Wright-Fisher simulations. Then we enumerated the set of DNA sequences that differed by one mutation from any of these binding sites, but were not themselves part of the focal transcription factor's landscape. Finally, we determined the fraction of transcription factors in our dataset that these one-mutant neighbors bound. This fraction was our measure of evolvability.

## Supporting information

**S1 Fig. Non-dominant genotype network components usually comprise few sequences.** Histogram of non-dominant component sizes. In total, 46% are singletons and 97% are not large enough to satisfy our inclusion criterion of containing 100 sequences. (TIF)

**S2 Fig. Composition bias is most pronounced in landscapes with low or high GC content.** The composition bias in entire landscapes, and in accessible mutational paths connecting the 10% of binding sites with the lowest affinity to the global peak, is shown in relation to the average GC content of the sequences in the landscape. Data pertain to all 746 landscapes. Notches indicate medians, whiskers indicate the 25th and 75th percentiles, and cross symbols indicate

outliers. The horizontal line indicates no composition bias (0.5).
(TIF)

**S3 Fig. Composition bias becomes more pronounced as the noise threshold parameter $\delta$ increases.** For each landscape and for five different values of the noise threshold $\delta$, we calculated the composition bias along the accessible mutational paths connecting the 10% of binding sites with the lowest affinity to the global peak. Data pertain to all 746 landscapes. Black dots indicate medians, whiskers indicate the 25th and 75th percentiles, and cross symbols indicate outliers.
(TIF)

**S4 Fig. Transversions cause larger changes in binding affinity than transitions, but only near the global peak.** The % increase in binding affinity conferred by transition and transversion mutations along accessible mutational paths is shown in relation to the mutational distance of a binding site to the global peak. For each binding site in each accessible path at each mutational distance $d$, we calculated the increase in affinity as the percentage change at mutational distance $d - 1$ along the path, relative to the affinity of the binding site at distance $d$. Notches indicate medians, whiskers indicate the 25th and 75th percentiles, and cross symbols indicate outliers. Mutational distances 1 and 2 exhibit statistically significant differences in the increase in binding affinity conferred by transitions and transversions (Bonferroni corrected two-sample $t$ test, $q < 10^{-3}$ and $q < 0.05$, respectively).
(TIF)

**S5 Fig. Constructing genotype-phenotype landscapes with *E*-scores or *Z*-scores results in highly similar composition biases among accessible mutational paths.** Pearson's correlation coefficient $r = 0.7212$, $p < 10^{-10}$. Data pertain to all 746 landscapes. The shaded gray regions highlight the 83 landscapes that switch from exhibiting a composition bias toward transversions to a composition bias toward transitions (or vice versa).
(TIF)

**S6 Fig. Quantifying composition bias using an alternative accessibility criteria results in very similar composition biases.** Data pertain to all 746 landscapes, Pearson's correlation coefficient($r = 0.8541$, $p < 10^{-12}$). The shaded gray regions highlight the 77 landscapes that switch from exhibiting a composition bias toward transversions to a composition bias toward transitions (or vice versa).
(TIF)

**S7 Fig. The mutation bias that maximizes $P_{peak}$ correlates strongly with composition bias, measured using two definitions of accessible mutational paths.** In **(a)**, each step in an accessible mutational path increases binding affinity by at least $\delta$. In **(b)**, each step in an accessible mutational path does not decrease binding affinity more than $\delta$. Data pertain to all 746 landscapes, each of which has its own noise threshold $\delta$.
(TIF)

**S8 Fig. Landscapes with strong composition bias comprise fewer binding sites than landscapes with little or no composition bias.** The number of binding sites per landscape is shown in relation to composition bias. Landscapes are grouped as in Fig 3. Data pertain to all 746 landscapes. Black dots indicate medians, whiskers indicate the 25th and 75th percentiles, and cross symbols indicate outliers.
(TIF)

**S9 Fig. Smaller landscapes are more navigable.** Landscapes are grouped based on the number of binding sites they comprise, with the average number of binding sites per landscape per

group ranging from 165.07 to 1037.60 in five linearly spaced increments. The probability of evolving to the global peak in the absence of mutation bias ($\alpha = 0.5$) is shown in relation to landscape size. Data pertain to all 746 landscapes. Black dots indicate medians, whiskers indicate the 25th and 75th percentiles, and cross symbols indicate outliers.
(TIF)

**S10 Fig. Mutation bias interacts with composition bias to influence landscape navigability, regardless of whether selection favors low or high affinity binding sites.** The probability $P_{\text{peak}}$ of reaching the global peak is shown for 19 different values of the mutation bias parameter $\alpha$. The solid vertical lines indicate no bias in mutation supply ($\alpha = 0.5$) and the dashed vertical lines indicate the value of $\alpha$ that maximizes $P_{\text{peak}}$. Landscapes are grouped based on their composition bias and the distribution of composition bias per panel is shown on top of each panel. The number of landscapes per panel is indicated is the bottom left corner. Fitness is a function of binding affinity (E-score) using the Gaussian function $\exp(-((E - E_{\text{opt}})/\sigma)^2)$, where E is the E-score of a binding site, $E_{\text{opt}}$ is the optimal E-score, and $\sigma$ is the variance parameter. Here, $E_{\text{opt}} = 0.35$ (the lowest E-score in our landscapes) and $\sigma = 0.1$.
(TIF)

**S11 Fig. Path entropy as a function of mutation bias.** Data pertain to 50 randomly chosen landscapes for 19 different values of the mutation bias parameter $\alpha$.
(TIF)

**S12 Fig. Mutation bias typically increases, but sometimes decreases, the predictability of evolution.** Shown are the percentage of mutation bias values $\alpha$ that increase or decrease path entropy (which is inversely related to the predictability of evolution), relative to when there is no mutation bias ($\alpha = 0.5$). Data pertain to all 19 values of the mutation bias parameter $\alpha$ on each of the 746 landscapes. Black dots indicate medians, whiskers indicate the 25th and 75th percentiles, and cross symbols indicate outliers.
(TIF)

**S13 Fig. Composition bias is a weak predictor of the mutation bias $\alpha$ that maximizes path entropy.** Nodes represent sequences in a landscape, and directed edges represent accessible mutations between sequences. Edge colors represent mutation type and node colors represent binding affinity (darker = higher). This landscape exhibits a strong composition bias toward transitions. Path entropy is therefore minimized by a strong mutation bias toward transversions, because an evolving population will utilize only one of the three accessible mutational paths. Conversely, one might expect path entropy to be maximized by a strong mutation bias toward transitions. However, this is not the case, because an evolving population will only utilize two of the three accessible mutational paths. The mutation bias that maximizes path entropy is actually the one that makes the three first-step mutations equiprobable. In more complex scenarios, with more and longer paths that include a greater diversity of binding affinities and more heterogeneous distributions of mutation types, a single summary statistic like composition bias is unlikely to accurately predict the value of mutation bias that maximizes path entropy.
(TIF)

**S14 Fig. Mutation bias has little to no effect on landscape navigability when the mutation supply is high.** The probability $P_{\text{peak}}$ of reaching the global peak is shown for 19 different values of the mutation bias parameter $\alpha$. This probability is calculated as the proportion of simulations in which at least half of the population evolves to the global peak. The solid vertical lines indicate no bias in mutation supply ($\alpha = 0.5$). Landscapes are grouped based on their

composition bias and the distribution of composition bias per panel is shown on top of each panel as in Fig 3.
(TIF)

**S15 Fig. Simulations of the Wright-Fisher model typically reach steady state within 1,000 generations. (a)** The average Shannon's diversity is shown in relation to the number of generations. After 827 generations, more than 99.9% of the 4, 252, 200 simulations reached steady state diversity levels within a tolerance of 0.01% of the final diversity level. The black line shows the average across all simulations. **(b)** The fraction of simulations that have reached steady state diversity levels is shown in relation to generation number.
(TIF)

**S16 Fig. Mutation bias influences the distribution of infinite populations on a genotype-phenotype landscape.** We characterized the steady state distribution of infinite populations as the eigenvector that corresponds to the largest eigenvalue of the matrix P (Methods). For any pair of such populations, we measured their overlap as the Euclidean distance between these eigenvectors—the shorter the distance, the higher the overlap. This panel shows this distance for pairs of populations in relation to the difference in their mutation bias parameters.
(TIF)

**S17 Fig. Mutation bias interacts with composition bias to influence the evolution of nucleotide diversity $\pi$. (a-e)** The final average nucleotide diversity of evolved populations at steady state is shown for 19 different values of the mutation bias parameter $\alpha$, and for each of three different values of mutation supply $N\mu$ (see legend). The solid vertical lines indicate no bias in mutation supply ($\alpha = 0.5$). Landscapes are grouped based on their composition bias and the distribution of composition bias per panel is shown on top of each panel, as in Fig 3.
(TIF)

**S18 Fig. Mutation bias and composition bias interact to influence allele frequency spectra.** The bars correspond to polymorphisms in the population at steady state, relative to the sequence that evolved to the highest frequency in the absence of mutation bias. Bar colors correspond to three different values of the mutation bias parameter $\alpha$: green for transversions ($\alpha = 0.05$), white for no bias ($\alpha = 0.5$) and blue for transitions ($\alpha = 0.95$). The panels correspond to two *Mus musculus* landscapes **(a)** Arid5a (composition bias toward transversions) and **(b)** Gm397 (composition bias toward transitions).
(TIF)

**S19 Fig. Genotypes in landscapes with strong composition bias are less robust than genotypes in landscapes without composition bias.** The y-axis shows the average mutational robustness of all genotypes in each landscape. The x-axis shows the composition bias. Landscapes are grouped as in Fig 3. Data pertain to all 746 landscapes. Black dots indicate medians, whiskers indicate the 25th and 75th percentiles, and cross symbols indicate outliers.
(TIF)

**S20 Fig. Mutation bias influences the evolvability of polymorphic populations.** The y-axis shows the difference in evolvability, which is calculated as the difference between the evolvability of a population at steady state when there is no mutation bias and when there is a strong bias toward transversions ($\alpha = 0.05$) or transitions ($\alpha = 0.95$). Data pertain to 128 transcription factors from **(a,b)** *Arabidopsis thaliana*, and **(c,d)** 128 transcription factors from *Mus musculus*. Landscapes are grouped according to their composition bias as in previous figures. Parameters: $N = 10^4$, $N\mu = 50$. As an example, for a given transcription factor, a 10% change in evolvability under strong transition bias could mean that the one-mutant neighbors of the

sequences evolved at steady state bind 11 transcription factors, whereas without mutation bias, the one-mutant neighbors of the sequences evolved at steady state bind 10 transcription factors.
(TIF)

**S1 Table. Information about the 746 transcription factors and their genotype-phenotype landscapes.**
(XLSX)

## Acknowledgments

We thank Arlin Stoltzfus for feedback on the bioRxiv preprint of this manuscript.

## Author Contributions

**Conceptualization:** Alejandro V. Cano, Joshua L. Payne.

**Data curation:** Alejandro V. Cano.

**Formal analysis:** Alejandro V. Cano.

**Funding acquisition:** Joshua L. Payne.

**Investigation:** Alejandro V. Cano.

**Methodology:** Alejandro V. Cano, Joshua L. Payne.

**Project administration:** Joshua L. Payne.

**Resources:** Joshua L. Payne.

**Software:** Alejandro V. Cano, Joshua L. Payne.

**Supervision:** Joshua L. Payne.

**Validation:** Alejandro V. Cano, Joshua L. Payne.

**Visualization:** Alejandro V. Cano, Joshua L. Payne.

**Writing – original draft:** Alejandro V. Cano.

**Writing – review & editing:** Alejandro V. Cano, Joshua L. Payne.

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
