## [Decision Letter · Decision Letter 0]

5 May 2020

Dear Prof. Payne,

Thank you very much for submitting your manuscript "Mutation bias interacts with composition bias to influence adaptive evolution" for consideration at PLOS Computational Biology.

As with all papers reviewed by the journal, your manuscript was reviewed by members of the editorial board and by two independent reviewers. In light of the reviews, below this letter, we would like to invite the resubmission of a revised version that takes into account the reviewers' comments.

We cannot make any decision about publication until we have seen the revised manuscript and your response to the reviewers' comments. Your revised manuscript is also likely to be sent to reviewers for further evaluation.

Sincerely,

Predrag Radivojac

Associate Editor

PLOS Computational Biology

Jian Ma

Deputy Editor

PLOS Computational Biology

Reviewer's Responses to Questions

**Comments to the Authors:**

Reviewer #1: In this submission, entitled "Mutation bias interacts with composition bias to influence adaptive evolution", Cano and Payne use protein-binding array data to investigate the interplay between the mutational bias and the composition of accessible sequences along the protein-binding landscape of transcription factor binding sites. The authors show that these two properties have implications for the probability that a population will ascend to the peak of this landscape (assuming that binding affinity is equated with fitness), the amount of diversity in the population, and the robustness of the population to the disruption of protein binding.

Overall I found this paper to be quite interesting and well supported. I believe that a couple of additional analyses could strengthen the authors' case, and I describe these below, followed by a list of minor comments primarily focused on parts of the manuscript that could be edited for clarity.

First, as the authors point out in the Discussion, binding affinity may not be equated with fitness (this should also be mentioned in the Introduction). I suggest that the authors experiment with creating a simple binding phenotype->fitness mapping, and seeing how this affects their results. For example, for their simulations they could adopt a Guassian fitness function, select an optimal value, and assign fitness values to each individual based on that individuals difference from the fitness optimum, and a variance parameter analogous to the strength of stabilizing selection. The authors could then experiment with different parameters of this fitness function, and see to what extent their conclusions about landscape navigability, population diversity, and robustness are affected.

Second, if I understand correctly, the authors define an accessible step along the binding landscape as one that improves binding affinity above some threshold. However, this seems overly conservative to me, as all regions of the landscape that do not substantially decrease fitness should be accessible--not every mutation along a path need be beneficial. Apologies if I misunderstood this, but otherwise I suggest that the authors repeat their experiments with a second definition of accessibility: any step that doesn't decrease binding affinity by at least some threshold (perhaps the same delta used by the authors) is accessible.

Finally, although I enjoyed the authors' analysis of the impact of mutation/composition bias on diversity, I think it would be helpful to include a couple of additional summaries. The authors could include nucleotide diversity (pi), which would paint a very similar picture to Shannon's diversity index but may be more familiar to some readers. I also would like to see them include allele frequency spectra, to give readers a picture of how mutation and composition bias may together influence the frequency of mutations in the population.

Minor comments:

In the intro, I would try to define composition bias more explicitly (i.e. the composition of sequences within the network/landscape, or all accessible sequences in the network in some cases).

Regarding path entroy: as far as I can tell path entropy is always zero or one in Figs 4a-b. Could the authors please explain this?

Top of page 7: I find it interesting that path entropy is decreased more by transversion bias than transition bias, given that in the absence of composition bias, transition mutation bias represents a greater constraint of available paths, as argued by the authors a few sentences above. Moreover, there is a transversion composition bias on average in accessible paths, which should again result in transition bias having a stronger impact on path entropy, right? Or perhaps I am misunderstanding. In either case, could the authors please explain/discuss in the text?

Also regarding path entropy: the value of alpha that maximizes path entropy is positively correlated with composition bias, as expected, but this seems to be a fairly weak correlation. Do the authors have any thoughts on why this is?

Figure 6: I recommend simulating more replicates for each mutation bias and landscape so that the resulting plots are less noisy--might help readers see the patterns more clearly. For example, in some cases there are dips near the edges of the x-axis, and this will make it clearer whether the presence/absence of such a dip is meaningful.

Regarding the section entitled "Mutation bias influences the evolvability of polymorphic populations": I found this section hard to understand in general, and in particular the last two sentences are kind of a word salad. I recommend a careful edit/rewrite pass for clarity.

I also think that "evolvability" needs to be defined more precisely, as "the ability of mutation to bring forth new binding phenotypes" is not specific enough in this context.

Finally, in that same section, the authors state that "S11 Fig shows how a bias in mutation supply affected the evolvability of polymorphic populations for 128 transcription factors from Arabidopsis thaliana and 128 transcription factors from Mus musculus, the two species with the most transcription factors in our dataset." I don't see how Fig S11 shows this. Perhaps the authors can clarify this in the legend.

Reviewer #2: Recent work on evolution on genotype networks motivated by computational genotype-phenotype models has changed how we view evolution in many ways. We now know that the networked structure of genotype space can alter evolutionary outcomes in ways that we couldn’t have predicted decades ago. An obvious advance in this topic is to study how mutation bias can modify this picture, by favoring some evolutionary pathways and thus affecting the chances to reach evolutionary optima. The effect of mutation bias is further compounded by composition bias, where a given landscape has an uneven proportion of nucleotides. The interaction between both biases and their effect on adaptive evolution is what the authors set to study in this paper, using a database of transcription factor-based landscapes.

The paper is well-written and relevant, and all in all I think it is more than suitable for PLoS Comp Biol. I have several minor comments that in my opinion would improve the manuscript, and also suggest further analysis and new methods in some cases: the authors are free to ignore these if they disagree with me or if what I ask is computationally expensive.

Minor comments:

1) P4, L120: “For some transcription factors (∼37%), the genotype network fragmented into several disconnected components. When this occurred, we only considered the largest component, which always comprised more than 100 bound sequences.”

I’m wondering, when there is more than one connected component, what is the distribution of their sizes? Is it always the case that one component is super large, while the others are very small? I guess if there are components of comparable size, the authors could consider them separately and independently: in the end, they are more or less parallel evolutionary universes somehow.

2) P5, L165: “Under these population genetic conditions, only one mutation is present in the population at any time [52], which makes the process amenable to modeling using Markov chains

(Methods).”

Not exactly. The fact that there is only one mutation at a time means we can model evolutionary dynamics as a random walk in genotype space.

This is just math nitpicking, but population dynamics can be modeled using Markov chains even in the presence of several mutations at the same time. For instance, the Wright-Fisher dynamics that the authors use later in the paper is in fact a Markov chain (it’s just that it is very impractical to use matrices in this case, and thus simulations are used). Markov processes appear whenever the probability of the process to jump to a given state just depends on the state the process is in now, and not on previous history. Most models of evolutionary dynamics are Markov chains.

3) In Fig3, I understand that the authors represent the average of P_peak across landscapes. I think showing the standard variation, or standard error would be informative. Is it always the case that the value of mutation bias that maximizes P_peak (for each landscape) is equal to the value of composition bias? If not, what is the average distance between these values?

4) In Fig4, I think showing some exemplars of alpha (x-axis) vs path entropy (y-axis) for a subset of landscapes would be very informative. How often is this function monotonous? What kind of shapes can we observe, and how are they related to composition bias? Is there any way to classify different landscapes according to this measure?

5) Also in Fig4b, it looks as if alpha_min is always either 0.05 or 0.95. Is that so?

6) In Fig5, there is (I believe) a better way to measure the overlap between steady-state populations, although I’m not sure if it’s computationally feasible, it will depend on the size of the networks.

Using Wright-Fisher simulations with finite-size populations will include a stochastic effect, that the authors have measured. However, if they want to test the topological effects of mutation bias, they could simulate the evolution of infinite populations using quasispecies dynamics. In that case, the steady-state distribution of a population on a genotype network is given by the eigenvector associated to the largest eigenvalue of the evolution matrix. See, for instance, section 3 in this paper (https://royalsocietypublishing.org/doi/full/10.1098/rsob.180069). The authors could just change matrix G in equation 3.5 to have G_ij=alpha if the connection is a transition and G_ij=1-alpha if it is a transversion, and then compute the eigenvector numerically: just multiply any initial vector by the transition matrix M, and then divide by the sum of the vectors components (so that the total sum is always one and so each component of the vector represents the probability that an infinite population is at that node), and keep doing this until the vector remains unchanged. That is the eigenvector associated with the largest eigenvalue (see also Box 1 in the same reference).

Now, this eigenvector will change with different values of alpha, and then any distance metric between vectors can be used to compare them.

This method is also valid for the results in Fig6.

7) In Fig6, the diversity measure used is not properly normalized. H_j (as defined in equation 9) can only equal one if every possible base can be visited by the population, but this is not the case, as not all nucleotide combinations are viable. Consider the landscape {AACG, CACG, TACG}. The maximum value of H_1 is log2(3) (which is greater than 1), while H_2, H_3 and H_4 will always be 0! Properly normalizing this measure in order to compare between networks is not easy.

A different (and in my mind, more natural way) to measure diversity would be to compute Shannon’s index over all genotypes. So H=\\sum p_i*\\log_2(p_i), where the sum ranges over all genotypes in the network. That is, p_i is the probability that the steady-state population is at a given node in the network (and this is given naturally by the eigenvector I mentioned in my previous point). The maximum value of H here is log_2(n), where n is the size of the network, so in order to compare between networks you just need to divide H by this value.

8) We computing mutational robustness of a steady-state population, is the average weighted with respect to the number of individuals in each genotype?

**Have all data underlying the figures and results presented in the manuscript been provided?**

Reviewer #1: No: All code essential for reproducing the authors' analyses should be publicly released.

Reviewer #2: Yes

PLOS authors have the option to publish the peer review history of their article (what does this mean?). If published, this will include your full peer review and any attached files.

Reviewer #1: No

Reviewer #2: Yes: Pablo Catalán
---

## [Decision Letter · Decision Letter 1]

30 Aug 2020

Dear Prof. Payne,

We are pleased to inform you that your manuscript 'Mutation bias interacts with composition bias to influence adaptive evolution' has been provisionally accepted for publication in PLOS Computational Biology.

Best regards,

Predrag Radivojac

Associate Editor

PLOS Computational Biology

Jian Ma

Deputy Editor

PLOS Computational Biology

Reviewer's Responses to Questions

**Comments to the Authors:**

Reviewer #1: The authors have done a thorough and satisfactory job addressing all of my comments, and I feel that the paper is now suitable for publication. I would also like to congratulate them on their grant.

Reviewer #2: The authors have answered all my queries. The manuscript is now more complete (maybe too complete!) and therefore I recommend it's publication as is.

**Have all data underlying the figures and results presented in the manuscript been provided?**

Reviewer #1: None

Reviewer #2: Yes

PLOS authors have the option to publish the peer review history of their article (what does this mean?). If published, this will include your full peer review and any attached files.

Reviewer #1: No

Reviewer #2: **Yes: **Pablo Catalán

---

## [Editor Report · Acceptance letter]

23 Sep 2020

PCOMPBIOL-D-20-00315R1 

Mutation bias interacts with composition bias to influence adaptive evolution

Dear Dr Payne,

I am pleased to inform you that your manuscript has been formally accepted for publication in PLOS Computational Biology. Your manuscript is now with our production department and you will be notified of the publication date in due course.

With kind regards,

Matt Lyles
